# Cost-effectiveness analysis of COVID-19 booster doses and oral antivirals: Case studies in the Indo-Pacific

**Gizem Mayis Bilgin**[1]*, **Syarifah Liza Munira**[2], **Kamalini Lokuge**[1], **Kathryn Glass**[1]

**1** National Centre for Epidemiology and Population Health, The Australian National University, Canberra, Australia, **2** Faculty of Economics and Business, Universitas Indonesia, Jakarta, Indonesia

* gizem.bilgin@anu.edu.au

**Data Availability Statement:** All data was collated from publicly available sources. Our model code and inputs can be viewed on GitHub: https://github.com/gizembilgin/indoPacific_COVID19_cost_effectiveness/tree/main. An interactive R Shiny

## Abstract

### Background

Decision-makers in middle-income countries need evidence on the cost-effectiveness of COVID-19 booster doses and oral antivirals to appropriately prioritise these healthcare interventions.

### Methods

We used a dynamic transmission model to assess the cost-effectiveness of COVID-19 booster doses and oral antivirals in Fiji, Indonesia, Papua New Guinea, and Timor-Leste. We conducted cost-effectiveness analysis from both healthcare and societal perspectives using data collated from publicly available sources. We developed an interactive R Shiny which allows the user to vary key model assumptions, such as the choice of discounting rate, and view how these assumptions affect model results.

### Findings

Booster doses were cost saving and therefore cost-effective in all four middle-income settings from both healthcare and societal perspectives using 3% discounting. Providing oral antivirals was cost-effective from a healthcare perspective if procured at a low generic price (US$25) or middle-income reference price (US$250); however, their cost-effectiveness was strongly influenced by rates of wastage or misuse, and the ongoing costs of care for patients hospitalised with COVID-19. The cost or wastage of rapid antigen tests did not appear strongly influential over the cost-effectiveness of oral antivirals in any of the four study settings.

### Conclusions

Our results support that COVID-19 booster programs are cost-effective in middle-income settings. Oral antivirals demonstrate the potential to be cost-effective if procured at or below a middle-income reference price of US$250 per schedule. Further research should quantify the rates of wastage or misuse of oral COVID-19 antivirals in middle-income settings.

**Funding:** The author(s) received no specific funding for this work.

**Competing interests:** The authors have declared that no competing interests exist.

## Introduction

Substantial population immunity has developed against coronavirus disease 2019 (COVID-19), with seroprevalence levels in 2023 above 90% in most countries due to immunity from previous infection or vaccination [1]. Three years after the emergence of COVID-19, vaccines supplies have been sufficient for primary schedule and partial booster dose coverage to reach low- and middle-income nations [2]. Still, older adults and individuals with comorbidities remain at considerable risk of severe outcomes associated with COVID-19 in 2023 [3, 4].

The World Health Organization (WHO) Strategic Advisory Group of Experts on Immunization (SAGE) roadmap for the use of COVID-19 vaccines (updated March 2023) urges policymakers in low- and middle-income countries to reevaluate the funding of COVID-19 interventions in the context of substantial population immunity against COVID-19 [3]. In a previous paper, we constructed a dynamic transmission model to estimate the impact of oral antivirals and booster doses on the burden of disease in Fiji, Indonesia, Timor-Leste, and Papua New Guinea. The model found that booster doses had the largest population-level impact in settings with high vaccine acceptance, and oral antivirals had the largest population-level impact in settings with lower vaccine acceptance [5]. This modelling provided evidence for the continued provision of booster doses to high-risk adults, and increased access to oral antivirals in middle-income settings. However, the feasibility of delivering further booster doses and/or oral antivirals will depend on their cost-effectiveness and relative opportunity cost compared to other healthcare priorities.

This paper examines whether COVID-19 booster doses and oral antivirals are cost-effective from healthcare and societal perspectives in Fiji, Indonesia, Papua New Guinea, and Timor-Leste. Understanding the relative efficiency of various interventions is crucial when deciding between competing healthcare priorities. This paper explores the cost-effectiveness of various combinations of booster dose and oral antiviral eligibility. We model an annual booster dose campaign with and without the year-round provision of oral antivirals to symptomatic adults at highest risk of severe COVID-19 outcomes. Highlighting trends across four middle-income nations with varying vaccine coverage, vaccine hesitancy, prevalence of comorbidities, and age structure extends the relevance of our findings to other middle-income settings. This paper seeks to answer (1) Whether annual booster programs are cost-effective in middle-income settings; (2) Whether oral antiviral programs have the potential to be cost-effective in middle-income settings; and (3) What factors are most influential over whether booster dose and oral antiviral programs are cost-effective in middle-income settings?

## Methods

### Study design

We evaluated the cost-effectiveness of COVID-19 booster dose and oral antiviral programs during 2023. We used GDP per capita as a default threshold for cost-effectiveness: Fiji (US $5,317), Indonesia (US$4,788), Papua New Guinea (US$3,020), Timor-Leste (US$2,358) [6]. We quantified the epidemiological impacts of COVID-19 booster doses and oral antivirals in a previous mathematical modelling paper [5]. In this paper, we estimated the cost of administering COVID-19 booster doses, the cost of delivering COVID-19 oral antivirals, and the costs averted by either intervention using publicly available data. Our comparator for this analysis was the continuation of existing primary and booster vaccine programs without oral antivirals and without further booster doses. We assumed the use of nirmatrelvir-ritonavir and provided additional results for molnupiravir in the S1 File and R Shiny. We present our analyses from both a societal and healthcare perspectives. A healthcare perspective includes only costs borne

by the health system; a societal perspective includes all costs regardless of who pays or benefits from them such as the cost of lost productivity due to illness and premature death. All prices were adjusted to 2022 United States Dollars using the International Monetary Fund's gross domestic product (GDP) deflators [7], as per [8].

We estimated the reduction of severe disease by booster doses and oral antivirals over the course of one year, aligning with existing economic evaluation of COVID-19 interventions [9]. We evaluated the impact of these interventions by including the ongoing costs and morbidity associated with cases of severe disease prevented within this year, discounting by 3%. Estimating the reduction in severe disease by oral antivirals over one year was appropriate because the effects of oral antivirals are short-term and localised to the individual receiving antivirals. Estimating the reduction in severe disease by booster doses over one year was conservative but appropriate because the increased protection against severe outcomes provided by booster doses compared to previous vaccination is expected to wane after one year [10].

### Health outcomes

**Underlying disease model.**    We have described the details of the deterministic COVID-19 transmission model and stochastic severe outcome projections in previous papers [5, 11], and a summary is provided in S1 Underlying Dynamic Transmission Model. In brief, we constructed a Susceptible-Exposed-Infected-Recovered (SEIR) model stratified by age, comorbidities, and vaccination status. This model was fit to the immunity profile of four study settings: Fiji, Indonesia, Papua New Guinea, and Timor-Leste (comparison of setting characteristics provided in S1.1 Table in S1 File). We modelled the provision of further booster doses and/or oral antivirals and estimated the number of doses needed to prevent one hospitalisation or death. Severe adverse outcomes associated with the use of vaccines or oral antivirals were not included in the model since there was insufficient documentation of these events in available literature.

**Estimation of health outcomes.**    We used the transmission model to estimate the health outcomes associated with different eligibility criteria for booster doses and oral antivirals. We considered scenarios without booster doses, with booster doses for high-risk adults, with booster doses for all adults, and with booster doses for adults who have received their primary schedule but not yet received a booster ('catch-up' campaigns). We assumed that oral antivirals would be given only to symptomatic adults at high risk of severe disease. We identified 'high-risk adults' as adults aged 18–59 with a known comorbidity and all adults aged over 60, as per current WHO Guidelines for delivery of COVID-19 antivirals and vaccines [3, 12]. We included additional scenarios for the provision of antivirals to unvaccinated adults in the S1 File since the WHO guidelines list an absence of COVID-19 vaccination as an additional risk factor to consider when dispensing oral antivirals [12].

**Translating health outcomes to QALYs.**    We calculated quality-adjusted life years (QALYs) saved by booster doses and/or oral antivirals by considering the number of deaths prevented, and incidence of nonfatal mild, severe, and critical COVID-19 averted. We adapted estimates by Robinson, Eber & Hammitt (2022) to translate the incidence of mild, severe, and critical COVID-19 to QALYs [13] (S2.1 Table in S1 File). This study by Robinson et al. considered the duration of symptoms and health-related quality of life (HRQoL) of nonfatal symptomatic cases. We converted deaths prevented into QALYs using life expectancy at each age from United Nations World Prospect Population estimates for 2022 [14], adjusted by the same age specific HRQoL estimates as in [13]. We considered QALYs associated with long COVID for additional sensitivity analyses presented in S4.7 of the S1 File.

**Table 1. Parameters informing booster dose and oral antiviral program costs.** All costs reported in 2022 United States Dollars. Probability distributions described where parameter values not fixed within simulations.

| Parameter | Component | Estimate | Source |
|---|---|---|---|
| Cost of vaccine | Price per dose | $1.50 –Johnson & Johnson<br>$0.70 –Moderna | UNICEF Supply Division COVID-19 vaccine price data [31] |
| | Wastage rate | 10%<br>Sensitivity analysis: 0%-50% | COVAX estimate for the expected wastage rate of COVID-19 vaccines [32], range informed by wastage rates of routine vaccines [18–20] |
| Injection equipment | Price per syringe and 1/100 safety box | $0.0351 | UNICEF Supply Division syringe and safety box bundles price data [31] |
| | Wastage rate | 10% | WHO guidelines for the introduction of new vaccines [33] |
| Operational costs (vaccine) | All other costs including cost of personnel, training, transport etc. | $2.85<br>Sensitivity analysis: $0.21-$13.04<br>Distribution: Lognormal (meanlog = 0.80, sdlog = 0.70) | Meta-analysis of routine immunisation delivery costs in low- and middle-income countries [17] |
| Cost of antiviral | Price per schedule | $530 high-income reference price<br>$250 middle-income reference price<br>$25 low generic cost | United States of America government agreement [26]<br>Malaysian government agreement [27]<br>Pfizer agreement for low- and middle-income countries, notably excluding Fiji [28] |
| | Wastage rate | 0%<br>Sensitivity analysis: 0–60% | Not considered in previous cost-effectiveness studies [34–35], range informed by an estimate for the proportion of overprescribed or inappropriately dispensed antimicrobials in middle-income countries [37] |
| Rapid antigen tests | Price per test | $2.225 | Median price across products listed on UNICEF catalogue [38] |
| | Wastage factor | 6<br>Sensitivity analysis: 3–12 | Estimate informed by distribution of influenza-like illness attributable to influenza [29] and likelihood of a COVID-19 positive individual testing positive on a rapid antigen test within the treatment window [30] |
| Operational costs (antiviral) | Price per outpatient visit for individual to be prescribed oral antivirals | $4.36 –Fiji<br>$2.57 –Indonesia<br>$3.91 –Papua New Guinea<br>$1.83 –Timor-Leste<br>Distributions provided in Table 2 | WHO CHOICE estimates for the cost per outpatient visit to a health centre with no beds [39] |

## Intervention costs

We used United Nations Children's Fund (UNICEF) Supply Division price data to estimate the price per dose of COVID-19 vaccines, syringes, and safety boxes (Table 1). All four study settings are eligible for the COVAX Advance Market Commitment [15]. We assumed that Indonesia, Timor-Leste, and Fiji used Moderna, and Papua New Guinea used Johnson & Johnson for future booster programs based on their previous vaccine supply as documented in UNICEF's Market Dashboard [16]. UNICEF prices for both vaccine doses and injection equipment are listed as free carrier prices, meaning that they include freight costs.

We estimated the operational cost per booster dose using a meta-analysis estimate of routine immunisation delivery costs in low- and middle-income countries [17]. Our sensitivity analysis was bounded by the minimum and maximum estimates of routine immunisation delivery costs included in this meta-analysis. Costs previously reported for supplementary immunisation activities and mass drug administration campaigns in our study settings ($0.33-$2.60 [18–21]) fell within the range in which we conducted sensitivity analysis. Previous routine immunisation programs for adults in our study settings have been limited to tetanus toxoid vaccination of pregnant mothers [22], and a recommendation for influenza vaccination in Indonesia for individuals intending to participate in a Hajj pilgrimage (<0.2% coverage) [23].

We used COVAX estimates for the wastage rates of COVID-19 vaccine doses. Data on country-specific COVID-19 vaccine wastage rates were not available [24]. We included

sensitivity analysis informed by the range of wastage rates reported for routine vaccines in our study settings [18–20]. WHO targets for acceptable wastage targets are 5% for double-dose vials and 15% for multi-dose vials in the third year of introduction [25].

We assessed the cost-effectiveness of oral antivirals at three schedule prices: a high-income country reference price of US$530 [26], a middle-income reference price of US$250 [27], and the anticipated low generic price of US$25 [28] for nirmatrelvir-ritonavir. We assumed that antiviral prices would be free carrier prices, as per previous agreements for the delivery of vaccine doses and injection equipment to low- and middle-income countries.

We estimated the number of rapid antigen tests (RATs) used for every schedule of oral antivirals (wastage factor) by considering the expected proportion of acute respiratory illnesses (ARI) attributable to COVID-19, and the likelihood of a case testing positive on a RAT. We assumed that the proportion of ARI attributable to COVID-19 would be similar to the proportion of influenza-like illness attributable to influenza prior to the introduction of COVID-19 (30%) [29] and included sensitivity analysis for lower (15%) and higher (60%) attributable fractions. Estimates from Menkir and colleagues informed the likelihood of a symptomatic case testing positive on a RAT within the treatment window (53.7%, 95% CI 27.1%-72.6%) [30]. Together, this data led to an estimate of six RATs (lower = 3, upper = 12) for every schedule of oral antivirals being dispensed. We assumed a home-based testing system where individuals tested at home before presenting to healthcare to receive oral antivirals.

## Healthcare and productivity costs averted

We calculated direct medical costs averted by preventing hospital admissions (inpatient costs), reducing the incidence of non-hospitalised symptomatic disease (outpatient costs), and reducing hospital length of stay (Table 2). Inpatient costs were estimated using the analysis by Nugraha et al. of COVID-19 inpatient costs recorded by the Indonesian Ministry of Health [40]. No primary data were available for our other three study settings. Instead, we adapted estimates from Indonesia to other study settings by fixing the costs of drugs and consumables, and adjusting the costs of healthcare workers' fees, accommodation, medical procedures, medical tests and examinations, and medical devices. These costs were adjusted by comparing WHO CHOICE estimates for the cost of inpatient bed days without drugs in international dollars between settings.

**Table 2. Parameters estimating healthcare costs associated with COVID-19.** All costs reported in 2022 United States Dollars. Lognormal distributions provided with meanlog and sdlog; normal distributions provided with mean and standard deviation.

| Component | Estimate | Sampling distribution | Source |
|---|---|---|---|
| Cost per hospital admission | $7,999.82 –Fiji<br>$5,847.48 –Indonesia<br>$3,858.55 –Papua New Guinea<br>$3,081.46 –Timor-Leste | Normal(7,999.82, 5.7)<br>Normal(5,847.48,4.1)<br>Normal(3,858.55,2.2)<br>Normal(3,081.46,2.2) | [40] |
| Cost per outpatient visit | $4.36 –Fiji<br>$2.57 –Indonesia<br>$3.91 –Papua New Guinea<br>$1.83 –Timor-Leste | Lognormal(1.24,0.689)<br>Lognormal(0.69,0.72)<br>Lognormal(0.68,1.13)<br>Lognormal(0.64,0.40) | [39] |
| Proportion of non-hospitalised symptomatic individuals accessing healthcare | 81.6%–Fiji<br>70.0%–Indonesia<br>70.0%–Papua New Guinea<br>76.6%–Timor-Leste | Fixed | [50–52] |
| Reduced length of stay of patients who received oral antivirals | 0.784 days | Normal(0.784,0.386) | [53] |
| Cost per additional inpatient day | 10% of cost per hospital admission | Normal(0.10,0.01) | [40] |

We made a conservative estimate for outpatient costs averted by using the lowest cost listed for outpatient care in WHO CHOICE estimates. Where available, we used setting-specific estimates on healthcare seeking behaviours of adults for mild respiratory illnesses. There were no data available for Papua New Guinea, so we assumed equal health seeking rates to Indonesia. Sensitivity analysis revealed that outpatient costs, and subsequently our estimates for health seeking rates, were not highly influential over the cost-effectiveness of either booster doses or oral antivirals.

We estimated healthcare savings due to the reduced length of stay of patients hospitalised after receiving oral antivirals. We did not estimate costs saved due to reduced lengths of stay of patients who had received additional booster doses. Evidence supports that vaccination reduces the length of stay in a general ward [41, 42] and in an intensive care unit (ICU) [43], but the effect of recent booster doses on length of stay have not yet been quantified.

We were not able to include direct non-medical costs in accessing healthcare—such as the cost of transportation, meals and care provided by family—due to an absence of data. A systematic review of paediatric infectious diseases in low- and middle-income countries found that direct non-medical costs of accessing care ranged between $1.21-$28.29 for inpatients and US$0-US$8.94 outpatients (2018) [44]. Hence the inclusion of this variable was unlikely to have a qualitative difference in the cost-effectiveness of either booster doses or oral antivirals from a societal perspective.

Productivity losses due to illness and premature death were estimated using a human capital approach—valuing lost time using an individual's gross earnings [45]. Productivity losses per outcomes, age group and setting are presented in S2.2 Table in S1 File. We estimated productivity losses due to premature death by considering years of life lost per age group and expected earnings in future years [14]. Expected earnings were estimated using setting- and age-specific labour force participation rates and annual outputs per worker modelled by the International Labour Organization [46]. We estimated productivity losses due to illness by considering Danish data on time taken to return to work after infection [47]; Danish data on return to work after COVID-19 infection was more granular and aligned with other available data form China and the Netherlands [48, 49].

## Sensitivity analysis to characterise uncertainty

We conducted both deterministic and probabilistic sensitivity analysis to understand how different model parameters influenced the cost-effectiveness of booster doses and oral antivirals. Deterministic sensitivity analysis—visualised in a tornado plot—was conducted by running the model with the lower and upper estimates of each parameter whilst keeping all other parameters constant. Probabilistic sensitivity analysis was conducted by running one thousand Monte Carlo simulations of the cost-effectiveness model randomly drawing from normal and lognormal distributions fit to the uncertainty of model parameters (S2.3 Table in S1 File). The corresponding one thousand incremental cost and one thousand incremental health effect values were visualised on an incremental cost-effectiveness plane to represent the uncertainty in our model. We also calculated the likelihood of interventions being cost-effective at varying willingness-to-pay thresholds using the one thousand Monte Carlo simulations. We have reported the probability of interventions being cost-effective with percentages in the results text in all cases where the estimated probability is not 100%.

## Access to interactive results and breakdown of parameter effects

We provide an interactive R Shiny for viewing and interacting with this paper's results on https://gizemmayisbilgin.shinyapps.io/indoPacific_COVID19_costEffectivenessAnalysis/ or

available directly for download https://github.com/gizembilgin/indoPacific_COVID19_cost_ effectiveness/tree/main/03_cost_effectiveness_analysis/07_shiny. The Shiny allows a user to vary key model assumptions, such as the discounting rate, and see how this affects the paper's results. The Shiny includes expected incremental cost-effectiveness ratios with 95% prediction intervals for all scenarios presented in Figs 1 and 2. A static overview of net outcomes and costs with corresponding incremental costs and benefits is provided in S3.1 and S3.2 Tables in S1 File. QALYs gained, intervention costs, healthcare costs averted, and productivity losses prevented by the source of these incremental differences is provided in S3.1 and S3.2 Figs in S1 File.

## Results

### Cost-effectiveness of booster doses

Annual booster programs were cost saving and therefore cost-effective from both healthcare and societal perspectives in all study settings and under all eligibility criteria (Fig 1). Programs which provided booster doses to all adults were estimated to be more cost-saving than booster programs which provided booster doses to high-risk adults only, particularly from a societal perspective. Differences in the incremental costs and benefits of catch-up campaigns varied by the vaccination history of the study setting. For example, a catch-up campaign in Indonesia saved a similar number of QALYs but prevented lower productivity costs than a campaign which provided booster doses to all adults. This was because prior booster coverage was higher in younger adults than older adults, so a catch-up campaign would provide more booster doses to older adults (group at highest risk of severe outcomes) and less doses to younger adults (group with highest productivity losses). By comparison, in Fiji, there was little difference in vaccine uptake by age, so a catch-up campaign saved a higher number of QALYs and prevented larger productivity costs than a campaign which provided booster doses to all adults.

Deterministic sensitivity analysis supported that providing booster doses to all adults would be cost saving and therefore cost-effective using the lower and upper estimates of all model parameters (S4.1 and S4.2 Figs in S1 File). The only exception was in Timor-Leste, where a booster dose program for all adults had an expected mean cost of $388 per QALY gained when using the upper bound estimate for the operational cost of delivering booster doses. This program would still be cost-effective using Timor-Leste's GDP per capita ($2,358) as a threshold.

The cost of caring for patients hospitalised due to COVID-19 had the largest influence over the cost-effectiveness of booster doses in most settings from a healthcare perspective (Fig S4.1 Fig in S1 File). Productivity losses due to illness had the largest influence on the cost-effectiveness of booster doses from a societal perspective (S4.2 Fig in S1 File). Booster wastage rates and the cost of injection equipment did not have a large influence over the cost-effectiveness of booster doses from either a societal or healthcare perspective. The order of the importance of other parameters—most notably, the operational cost of delivering booster doses—varied across settings.

### Cost-effectiveness of oral antivirals

Oral antiviral programs for high-risk adults demonstrated the potential to be cost-effective in all settings using Pfizer's agreed low generic price, were cost-effective in three settings (not Timor-Leste, only 41% likelihood) using the middle-income reference price, and in one setting (Papua New Guina with 61% likelihood) using the high-income reference price from a healthcare perspective with GDP per capita as a threshold for cost-effectiveness (Fig 2). In comparison, from a societal perspective, providing oral antivirals to high-risk adults was cost saving in

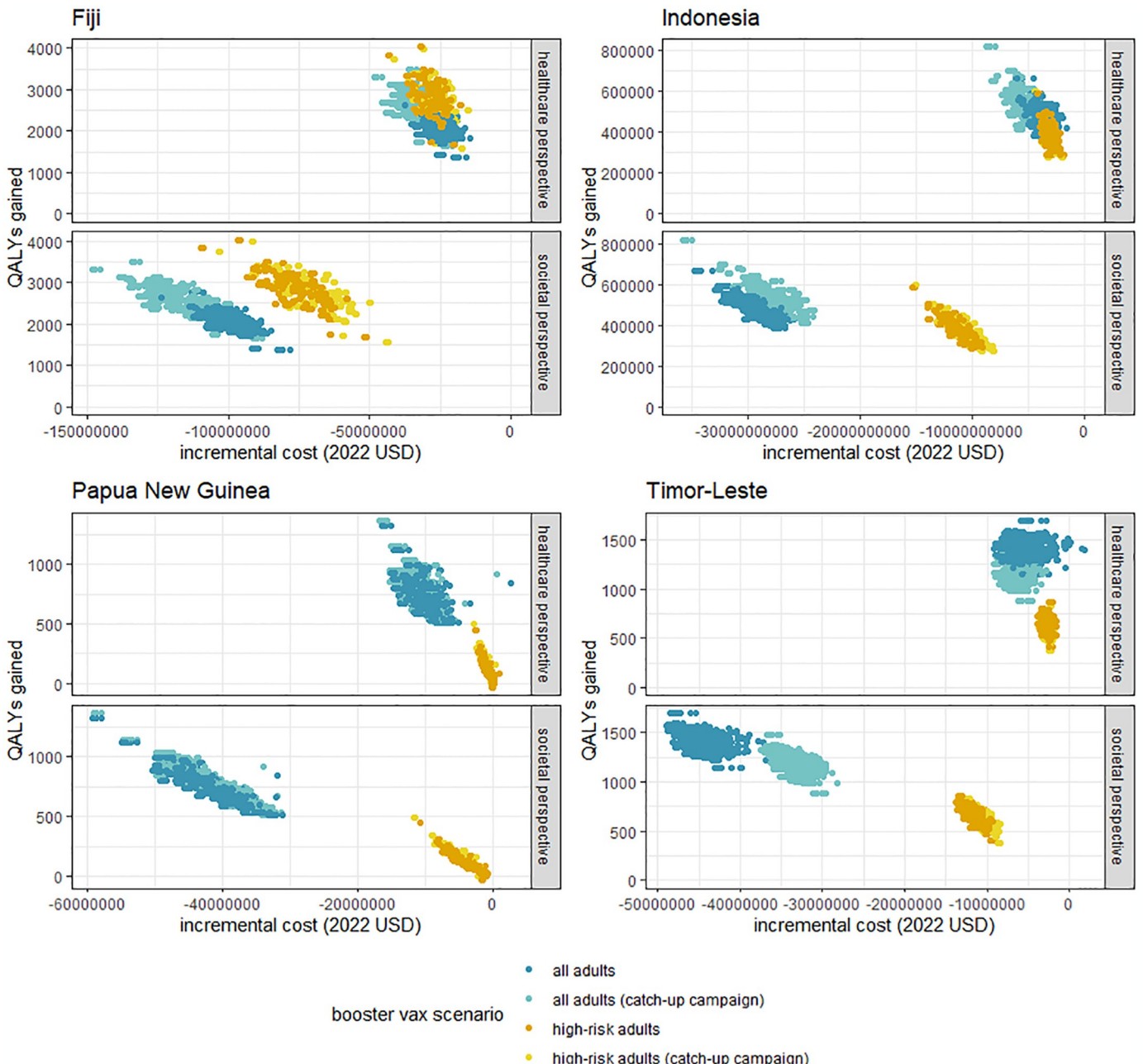

**Fig 1. Comparison of the incremental benefits (QALYs) and incremental costs of booster doses in 2023.** Results are presented from both healthcare and societal perspectives using 3% discounting of ongoing health benefits and assuming no oral antivirals are available in the study settings. Each point represents one simulation. One thousand Monte Carlo simulations are represented for each booster eligibility scenario and choice of perspective.

three settings (not Timor-Leste), and cost-effective in all settings even when using the high-income reference price (53% likelihood in Timor-Leste and 100% in three other settings) (S4.5 Fig in S1 File).

The cost of oral antivirals per schedule, antiviral wastage rates, and the cost of care for patients hospitalised due to COVID-19 were the three primary drivers of whether oral antivirals were cost-effective from a healthcare perspective (S4.3 Fig in S1 File). Productivity losses

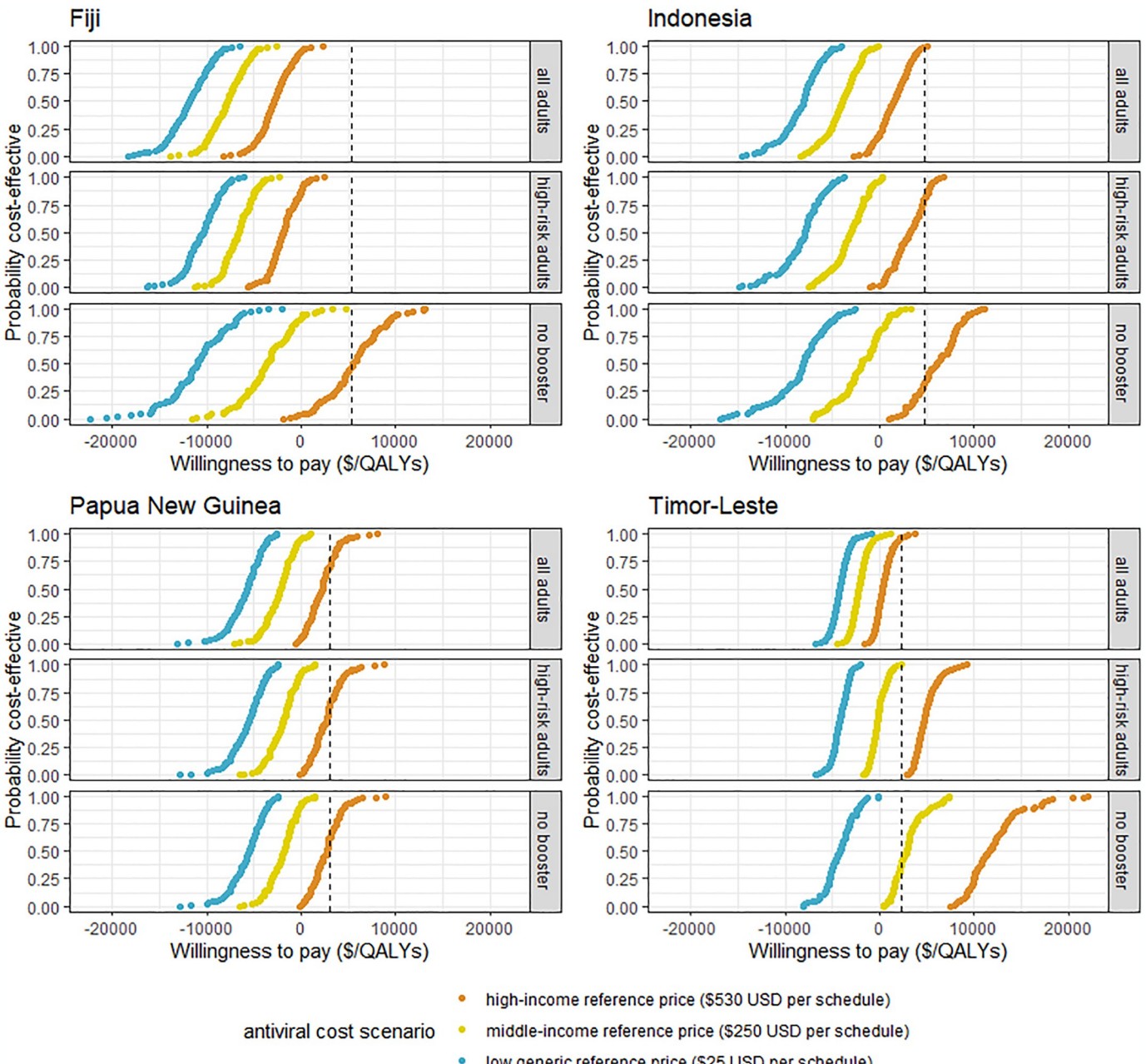

**Fig 2. Comparison of the probability of an oral antiviral being cost-effective by willingness to pay United States Dollars per QALY across three antiviral schedule prices and three booster eligibility strategies from a healthcare perspective.** Probabilities have been calculated as a % of one thousand Monte Carlo simulations. All scenarios assume that nirmatrelvir-ritonavir type oral antivirals are provided to symptomatic high-risk adults and use 3% discounting of ongoing health benefits. The dashed line represents the nation's gross domestic product per capita as a possible threshold for cost-effectiveness.

due to illness had a larger influence than productivity losses due to death over the cost-effectiveness of oral antivirals from a societal perspective (S4.4 Fig in S1 File). Interestingly, the price or wastage of RATs did not appear strongly influential over the cost-effectiveness of oral antivirals in any of the four study settings. If the wastage of oral antivirals was high, oral

antivirals could only be cost-effective if procured at the low generic price in all study settings from a healthcare perspective (S4.8 Fig in S1 File).

Oral antiviral programs for unvaccinated adults would only be cost-effective if procured at the low generic price from a healthcare perspective (S4.6 Fig in S1 File). Indonesia was the only setting where an oral antiviral program for unvaccinated individuals would be cost-effective if purchased at the middle-income reference price. This was due to Indonesia's relative high GDP per capita and lower vaccine coverage in older adults. Interestingly, providing oral antivirals to all symptomatic adults would be cost-effective if a large supply of oral antivirals was available at the low generic reference price (S4.6 Fig in S1 File). An oral antiviral with lower effectiveness, as seen with molnupiravir, would only be cost-effective from a healthcare perspective if the antiviral were available at the low generic cost (S4.7 Fig in S1 File).

## Cost-effectiveness of oral antiviral—Booster dose combinations

Providing booster doses to all adults and oral antivirals to high-risk adults appeared cost-effective in all study settings from both a healthcare and societal perspective at all antiviral schedule prices using GDP per capita as a cost-effectiveness threshold (Fig 2). Further, this combination strategy appeared cost saving from a healthcare perspective using either the low-generic or middle-income reference prices for oral antivirals, and cost saving from a societal perspective using all antiviral reference prices. The three parameters which had the largest influence over the cost-effectiveness of a combination strategy from a healthcare perspective were the cost of oral antivirals per schedule, antiviral wastage rates, and the cost of care for patients hospitalised due to COVID-19 –i.e., the same parameters which had the largest influence over the cost-effectiveness of a stand-alone oral antiviral program (Fig 3).

## Discussion

Annual COVID-19 booster programs demonstrate a strong potential to be cost-effective in middle-income settings. Providing oral antivirals to high-risk adults is also likely to be cost-effective from a healthcare perspective if antivirals can be procured at appropriate prices. This strategy is certainly cost-effective at Pfizer's agreed low generic price, and highly likely to be cost-effective at our middle-income reference price (Malaysia's negotiated price). The price or wastage of RATs was not influential over the cost-effectiveness of an oral antiviral program; however, the wastage rate of oral antivirals was of concern from a healthcare perspective. The choice of discounting did not affect whether an intervention was cost-effective. Most interventions were cost-effective from a societal perspective.

Findings on the cost-effectiveness of booster doses and oral antivirals were fairly consistent across the four study settings. The impact of different booster dose and antiviral programs on the burden of disease varied across study settings due to prior vaccine coverage, prevalence of comorbidities, and age-structure; however, qualitative decisions on which eligibility strategies were cost-effective and the order of parameter importance in dictating cost-effectiveness were largely consistent between study settings. Timor-Leste was least similar to the three other settings due to its lower GDP per capita, relatively young population, and lower prevalence of comorbidities.

Our analysis suggests that the cost-effectiveness of oral antivirals may depend on the rates of antiviral wastage or misuse. Wastage rates for antivirals will include courses dispensed which are unused, courses dispensed for individuals who do not have COVID-19 (false positives and patients with other respiratory illnesses), courses which expire before prescription, and courses dispensed too late to be effective. Neither high- nor middle-income countries have publicly released rates of COVID-19 oral antiviral wastage or misuse at the time of this

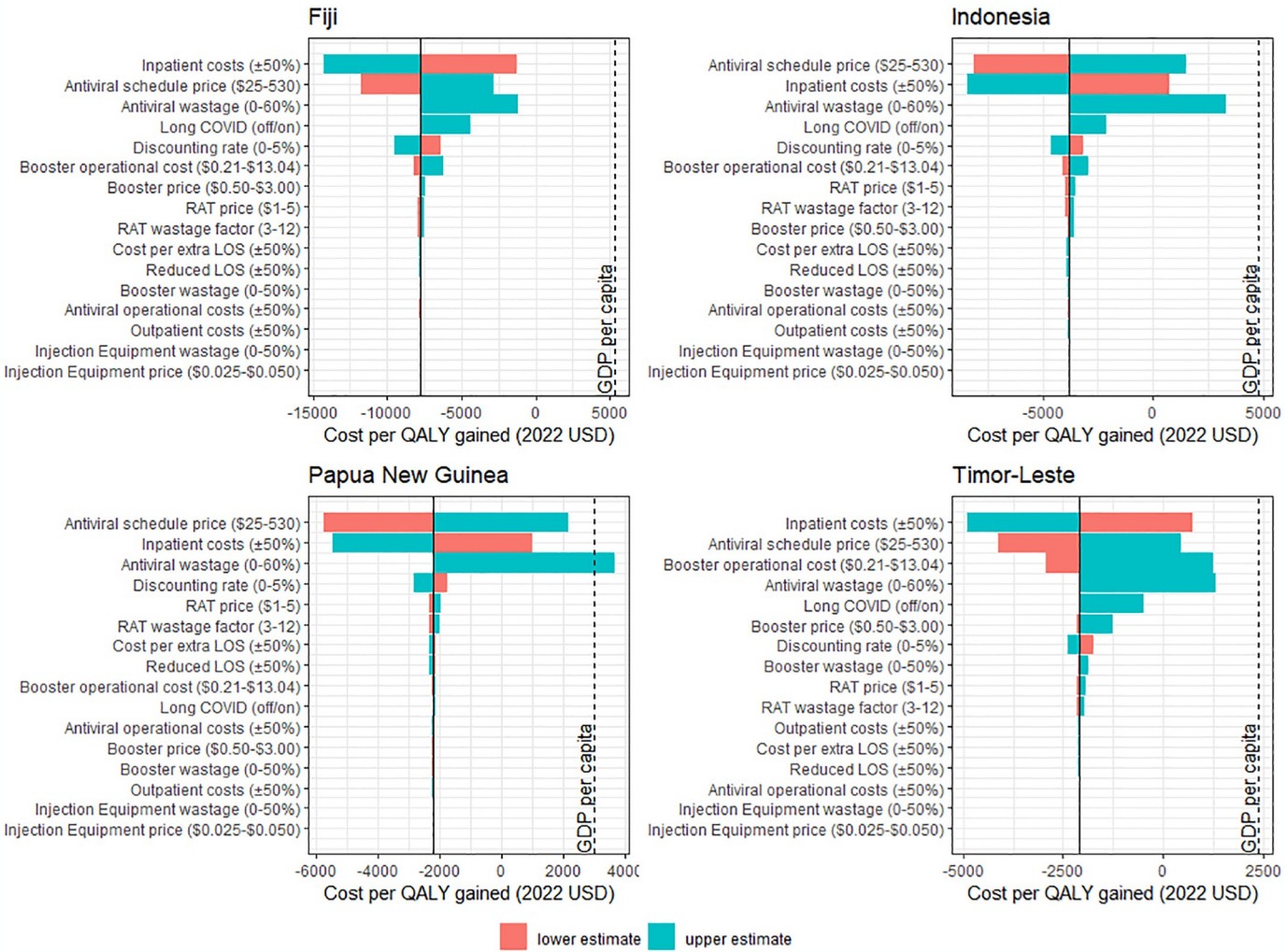

**Fig 3. Tornado plot visualising one-way deterministic sensitivity analysis of the cost-effectiveness of providing booster doses to all adults and oral antivirals to high-risk adults in 2023 from a healthcare perspective.** Individual deterministic sensitivity analysis for the effects of providing booster doses to all adults and oral antivirals to high-risk adults are provided in S4.1 and S4.3 Figs in S1 File respectively. The dashed line represents the nation's gross domestic product (GDP) per capita as a possible threshold for cost-effectiveness.

study. This may be due to the political sensitivity surrounding this data, challenges in collecting these data and/or the relative novelty of oral antivirals in these settings. It will be important to measure and monitor the rates of antiviral wastage when oral antiviral programs are implemented in middle-income countries.

Negotiations on appropriate oral antiviral prices for middle-income countries will be key in their cost-effectiveness. The WHO believes that a lack of price transparency in deals with pharmaceutical companies has made accessing oral antivirals a major challenge for low- and middle-income countries [54]. We used three reference prices throughout our analysis—a low generic price based on Pfizer's commitment [28], a middle-income reference price based on Malaysian government documentation [27], and a high-income reference price based on the United States of America's publicly released agreement [26]. Our results demonstrated that oral antivirals will be cost-effective if procured at a low generic or middle-income reference prices. Of our study settings, Indonesia, Papua New Guinea and Timor-Leste (not Fiji) are

eligible for Pfizer's low price generics [55]. Pooled procurement mechanisms have already achieved price reductions of 30–50% for diagnostic tests and between 10–50% on oxygen for low- and middle-income countries over the course of the pandemic [56]. Further, 38 generic manufacturers in 13 countries– 11 being low- and middle-income countries—have been signed for nirmatrelvir-ritonavir [56]. If production is implemented at scale, there should be sufficient supply to cover the needs of low- and middle-income countries. Pooled procurement mechanisms and the scaling up of generic manufacturing should enable oral antivirals to be accessed at cost-effective prices in middle-income settings.

Our results are comparable to similar studies, although there have been a limited number of studies published on the cost-effectiveness of oral antivirals or booster doses in low- and middle-income countries. A recent systematic review on the cost-effectiveness of COVID-19 vaccines noted a gap in published literature on the cost-effectiveness of booster doses (June 2023) [57]. Our results align with available studies which demonstrate that booster doses appear cost saving in middle and high-income settings [58–60]. Previous studies have demonstrated that providing nirmatrelvir-ritonavir to high-risk adults would be cost-effective in high-income settings [35, 61, 62]. We identified only one study considering the cost-effectiveness of oral antivirals in low- or middle-income countries (in preprint [63]). This study demonstrated that nirmatrelvir-ritonavir was cost-saving when provided to older adults in Rwanda, Zambia, and Ghana at Pfizer's low generic price. The study identified the price of antivirals and the costs of hospitalisation as key determinants of the cost-effectiveness of oral antivirals, aligning with our deterministic sensitivity analysis. Future work should consider extending the wealth of modelling on epidemiological impact to also include an assessment of the cost-effectiveness of oral antivirals and booster doses. This will better allow decision-makers to compare these interventions against other healthcare priorities.

A key limitation in the applicability of our findings is uncertainty in pharmaceutical supplies and ongoing health workforce capacity to deliver COVID-19 interventions. Theoretically, providing booster doses to all adults and oral antivirals to all symptomatic adults appeared cost saving in all study settings from a healthcare perspective if oral antivirals were procured at Pfizer's low generic price. However, such a large program would require sufficient supplies of pharmaceuticals, appropriate storage capacity, dedicated health workforce capacity, and investment in surveillance programs to monitor vaccine effectiveness and antiviral resistance. Three years into the pandemic, decision-makers have begun to compare the opportunity cost of investing in COVID-19 interventions against other healthcare priorities [3].

Our study has several limitations due to the availability or time-dependence of data informing model parameters. We used GDP per capita as a default threshold for cost-effectiveness. The limitations of this approach have been clearly discussed in literature [64–66]. Cost-effectiveness analyses of individual interventions are not representative of real-world decision-making scenarios. Contextual factors such as competing financial demands, cultural values and healthcare capacity will determine a decision maker's willingness to pay per QALY saved by a COVID-19 intervention. Further, if COVID-19 programs transition to private funding than analysis will need to be conducted from a healthcare payer perspective and compared to consumer willingness to pay thresholds (as measured in [67, 68]). The cost of inpatient care had a considerable influence on the cost-effectiveness of both booster doses and oral antivirals in our analysis. Estimates for the cost of caring for patients hospitalised with COVID-19 were from data collected during 2020 and 2021 [40]. Changing standards of care, such as reduced requirements for isolation, may reduce these costs over time. Productivity losses due to illness had a substantial impact on the cost-effectiveness of both interventions from a societal perspective. We used estimates on the time taken to return to work after COVID-19 from a Danish study since no data were available from comparable middle-income contexts. Delays in

returning to work may be longer (lower access to healthcare) or shorter (greater financial need) in middle-income settings. Hence, the characterisation of this parameter should be prioritised if cost-effectiveness analysis is to be conducted from a societal perspective. There remains limited data on the ongoing healthcare costs and productivity losses associated with long COVID, hence, we were unable to include long COVID in our main analysis. Cost-effectiveness analyses of COVID-19 booster doses and oral antivirals should be updated as timelier, and more setting-relevant data emerges.

## Conclusions

Our study supports the continued provision of booster doses as the most cost-effective intervention against COVID-19. Our results demonstrate the potential for oral antivirals to also be cost-effective if procured at reasonable prices for middle-income settings and implemented with low wastage. Future rollout of both interventions should monitor their wastage rates, the cost of care for patients hospitalised with COVID-19, and emerging evidence on the impacts of long COVID. The relative cost-effectiveness of investing in COVID-19 booster doses and oral antivirals compared to the opportunity costs of investing in other healthcare interventions will vary as the levels of transmission and severity of COVID-19 change over time.

## Supporting information

**S1 File. Supplementary material.** Additional detail on the underlying dynamic transmission model (S1), cost-effectiveness parameter estimates (S2), a breakdown of the results (S3), and results from additional scenario analysis (S4).
(PDF)

## Author Contributions

**Conceptualization:** Gizem Mayis Bilgin, Syarifah Liza Munira, Kamalini Lokuge, Kathryn Glass.

**Formal analysis:** Gizem Mayis Bilgin.

**Investigation:** Gizem Mayis Bilgin.

**Methodology:** Gizem Mayis Bilgin, Syarifah Liza Munira.

**Software:** Gizem Mayis Bilgin.

**Validation:** Kathryn Glass.

**Visualization:** Gizem Mayis Bilgin.

**Writing – original draft:** Gizem Mayis Bilgin.

**Writing – review & editing:** Gizem Mayis Bilgin, Syarifah Liza Munira, Kamalini Lokuge, Kathryn Glass.

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
