## [Decision Letter · Decision Letter 0]

26 Dec 2023

PONE-D-23-34679Cost-effectiveness analysis of COVID-19 booster doses and oral antivirals in the Indo-PacificPLOS ONE

Dear Dr. Bilgin,

Thank you for submitting your manuscript to PLOS ONE. After careful consideration, we feel that it has merit but does not fully meet PLOS ONE’s publication criteria as it currently stands. Therefore, we invite you to submit a revised version of the manuscript that addresses the points raised during the review process.

We look forward to receiving your revised manuscript.

Kind regards,

Harapan Harapan, MD, PhD

Academic Editor

PLOS ONE

Journal Requirements:

3. Please ensure that you refer to Figure 3 in your text as, if accepted, production will need this reference to link the reader to the figure.

4. We note you have included a table to which you do not refer in the text of your manuscript. Please ensure that you refer to Table 3 in your text; if accepted, production will need this reference to link the reader to the Table.

Reviewers' comments:

Reviewer's Responses to Questions

**Comments to the Author**

1. Is the manuscript technically sound, and do the data support the conclusions?

Reviewer #1: Partly

Reviewer #2: Partly

2. Has the statistical analysis been performed appropriately and rigorously? 

Reviewer #1: I Don't Know

Reviewer #2: No

3. Have the authors made all data underlying the findings in their manuscript fully available?

Reviewer #1: Yes

Reviewer #2: Yes

4. Is the manuscript presented in an intelligible fashion and written in standard English?

Reviewer #1: No

Reviewer #2: Yes

5. Review Comments to the Author

Reviewer #1: Thank you for the opportunity. Please find below my comments:

[Abstract]

1. State how or where the data were derived from.

2. Authors stated that the findings support the government program to be cost-effective. This statement sounds like the government make the same cost-effective analysis and the authors compared their findings with that of government's. I believe this is due to writing technicalities. Authors should revise the sentence so that it won't sound that as if the government make the same study.

[Introduction]

1.Line 47-48. Data about COVID-19 mortality and morbidity are fluctuative since the disease is newly emerged. Please state when the data was recorded.

2. Authors should provide why it is important to perform cost-effective study, this include its correlation with vaccine acceptance level. (for example, authors may incorporate findings from this study: Narra J. 2022. 2(2): e85)

3. Other than being previously studied (by the same authors), justification on choosing Fiji, Indonesia, etc. should be provided in introduction.

[Methods]

1. What do authors mean by societal and healthcare perspectives?

2. Please state how the data were collected in the study design.

3. Line 140-142. It is not clear how the assumption was made for the future vaccine products of choice.

[Results]

1. Focus the paragraph to present the data -- not repeating the methods.

[Discussion]

1. Line 380-381. Other than being politically sensitive, some of the countries might suffer from their inability to collect proper data (absence of electronic records for instance or low survaillance for drug misuse).

2.Authors mentioned about nirmatrelvir-ritonavir, but before this drugs there have been several antivirals used to treat SARS-CoV-2, please discuss them as well, and how the use of those drugs implicate the present study. (author may incorporate this study: Narra J. 2022. 2(3):e92)

[Conclusion]

1. Your findings support the continuation of gov-funded COVID-19 vaccination and procurement of oral anti-SARS-CoV-2. Please make clear about the assumption used in the study, and how the findings are not generalizable. For example, data about COVID-19 fatality and transmission is keep changing through times (if not months). Will your study still relevant if the fatality/transmission rate changes?

Reviewer #2: I would like to congratulate the authors for this important work. This study gives a good insight into estimation of the cost-effectiveness related to covid-19 vaccine. The authors have done a great work and stated all the analytical assumption behind the outputs. However, there are some major issues that the authors should address:

Major comment

1. The use of QALY derived from VSL which estimated the value of 1 QALY to $580,000. This means the willingness to pay/threshold should be more than $580,000 as per the study by Robinson LA. Estimating the QALY gained by averting mild, moderate, or severe cases of COVID should be done first by a valid HRQOL instrument like EQ5D (https://euroqol.org/support/analysis-tools/index-value-set-calculators/) and then changed to utility score using societal valuation (tariff) for the specific countries. Therefore, the authors need to clarify why they used estimates done by Robinson LA. If this is justified, the study needs alternative analysis method to estimate QALYs.

2. Supplementary file table 2 also mention a huge difference in QALY for mild, severe, fatal cases and critical cases. The QALY values have a huge gap, perhaps due to considering future health states for 2 of the health outcomes and not considering future health states for the other outcomes. Therefore, the authors need to provide the formula on how each of the QALYs for the mild, critical, severe, and fatal condition are calculated.

3. Line 89-95 time horizon of one year is used, if so we do not need to discount using 3%. However, the time horizon looks life time when the supplementary table 2 is seen.

Minor comments

1. The report is well written in detail but it needs restructuring it. To structure the report, please use CHEERS 2022 guideline as it can be a guide for the authors on how to structure this report. this will make it easy for the readers to follow.

2. The paper needs to be corrected for some typos.

3. The result section needs to have numbers reported in it. Example: when doing probabilistic sensitivity analysis, the authors should report the probability of being cost-effective with percentage.

4. In the method section, we need to include the threshold that is used for each country (GDP per capita) and explain why it is preferred.

Line by line comments

Abstract: line 3 repeated statement of 3% discount rate.

Abstract: Line 29, 30 A low generic. Do you mean generic price for low-income countries? Please, correct the sentence.

Abstract Line 32 Please, do not use subjective terms like “Interestingly”.

Abstract Line 37: I would recommend not using “continue to be”, as this study will make it look the authors are doing this study for the second time.

Abstract Line 38: “$250 USD”; please correct this to” US$ 250”

Line 59-63: three of the countries are classified as middle-income. Page 4 line 69-73: informs this estimation is for middle-income countries. I recommend the generalizability from this study for middle-income countries, not low-and middle-income countries.

Line 89-95: For a time horizon of one year, we do not need to discount the cost and effect. Discounting will only be applicable for cost and effects that lasts more than one year.

Line 146-47: I suggest all reference estimates to represent middle-income countries only to estimate the operational cost.

Please, write the title of all the figures

Figure 1: please correct figure 1 vertical axis to “incremental QALY gained”, not “QALY averted”

Figure 2: I would suggest not using negative willingness to pay threshold.

Again correct the QALY averted to QALY gained.

Supplementary table 2: in Fiji for age group 0 to 4; QALY= 0.01 for mild, 0.021 for severe, 4.884 for critical cases, and 26.6 for fatal cases.

The above figure looks the QALYs gained for mild, severe, critical, and fatal cases were calculated in different ways. I can see that the QALY for mild and severe are calculated without considering health gains. However, the QALY for critical and fatal cases are calculated using 3% discount rate including the future health gains. Therefore, I strongly recommend for the authors to do the analysis again with the support of senior health economist. In addition, please note that QALY of 1 means the person is in a prefect health and QALY of 0 means the person is dead in a year per person. The authors also need to correct the words “QALY loss” to QALY gain.

6. PLOS authors have the option to publish the peer review history of their article (what does this mean?). If published, this will include your full peer review and any attached files.

Reviewer #1: No

Reviewer #2: **Yes: **Amanuel Yigezu

---

## [Author Response · Author response to Decision Letter 0]

24 Jan 2024

We thank the reviewers for their detailed comments. We have clarified sections of our paper and included additional detail based on these comments. These changes are detailed in our Response to Reviewer's document.

As per the editor's request, we have also checked the manuscript against PLOS ONE's style requirements, and ensured all figures and tables are referenced within the text.

---

## [Decision Letter · Decision Letter 1]

25 Jun 2024

PONE-D-23-34679R1Cost-effectiveness analysis of COVID-19 booster doses and oral antivirals in the Indo-PacificPLOS ONE

Dear Dr. Bilgin,

Thank you for submitting your manuscript to PLOS ONE. After careful consideration, we feel that it has merit but does not fully meet PLOS ONE’s publication criteria as it currently stands. Therefore, we invite you to submit a revised version of the manuscript that addresses the points raised during the review process.

We look forward to receiving your revised manuscript.

Kind regards,

Dominic Luke Thorrington, PhD

Academic Editor

PLOS ONE

Journal Requirements:

Additional Editor Comments:

The reviewers have highlighted some minor points in the manuscript that still require clarifications and further explanation.

Further information is required on the data that were retrieved from public sources - the reviewers are not convinced that this is sufficient for other researchers to be able to replicate the analysis. In addition, some clarifications from reviewer 2 concerning the presentation of methods and results is important to address.

Please carefully consider all of the points raised by both reviewers.

Reviewers' comments:

Reviewer's Responses to Questions

**Comments to the Author**

1. If the authors have adequately addressed your comments raised in a previous round of review and you feel that this manuscript is now acceptable for publication, you may indicate that here to bypass the “Comments to the Author” section, enter your conflict of interest statement in the “Confidential to Editor” section, and submit your "Accept" recommendation.

Reviewer #1: All comments have been addressed

Reviewer #2: (No Response)

2. Is the manuscript technically sound, and do the data support the conclusions?

Reviewer #1: Partly

Reviewer #2: Partly

3. Has the statistical analysis been performed appropriately and rigorously? 

Reviewer #1: Yes

Reviewer #2: I Don't Know

4. Have the authors made all data underlying the findings in their manuscript fully available?

Reviewer #1: Yes

Reviewer #2: Yes

5. Is the manuscript presented in an intelligible fashion and written in standard English?

Reviewer #1: Yes

Reviewer #2: Yes

6. Review Comments to the Author

Reviewer #1: Thank you for addressing my previous concerns. However, authors stated that the data were retrieved from 'publicly available sources'. This is not reproducible and convincing. Authors should provide more detail and at least use several criteria for the data. Some countries might not appropriately collect their data, hence doubt on the methodology.

Reviewer #2: Please see some comments below.

1. Line 95-99: Response not addressed properly from previous comment 3. The authors are not providing a clear response. The authors should either do the analysis with 1 year timeframe without considering future morbidity (consequences in terms both cost and outcome) or multiple year by discounting future consequences (for both the cost and outcome). In economic evaluation, It is methodologically incorrect to consider one effect from life time horizon and the cost from a one year horizon.

2. Table 2 parameter estimation “Cost per extra day hospital length of stay” is not clear.

3. Probabilities that are used in the model are not described in table. Please provide the probabilities that are used to populate the SEIR model, all parameters used to populate the model should be presented on one table.

4. Line 254: please review the distribution used for the parameters. “Commonly used distributions In economic modelling are symmetrical, such as the normal distribution, often used for parameters such as population age and intervention effectiveness (e.g. relative risk reduction), or skewed, such as gamma or lognormal, for ratios or for parameters such as costs which cannot be negative.”

5. Please, explain the results with numbers. See my previous comment from last review.

“The result section needs to have numbers reported in it. Example: when doing probabilistic sensitivity analysis, the authors should report the probability of being cost-effective with percentage.”

7. PLOS authors have the option to publish the peer review history of their article (what does this mean?). If published, this will include your full peer review and any attached files.

Reviewer #1: No

Reviewer #2: **Yes: **Amanuel Yigezu

---

## [Author Response · Author response to Decision Letter 1]

1 Jul 2024

Please see our response to reviewers letter

---

## [Decision Letter · Decision Letter 2]

30 Jul 2024

PONE-D-23-34679R2Cost-effectiveness analysis of COVID-19 booster doses and oral antivirals in the Indo-PacificPLOS ONE

Dear Dr. Bilgin,

Thank you for submitting your manuscript to PLOS ONE. After careful consideration, we feel that it has merit but does not fully meet PLOS ONE’s publication criteria as it currently stands. Therefore, we invite you to submit a revised version of the manuscript that addresses the points raised during the review process.

We look forward to receiving your revised manuscript.

Kind regards,

Dominic Luke Thorrington, PhD

Academic Editor

PLOS ONE

Journal Requirements:

Additional Editor Comments:

The latest round of review has resulted in a recommendation for further minor revisions, mainly some methodological clarifications and addressing some small short-comings. Please read the reviewer's response carefully to address all of their comments.

Reviewers' comments:

Reviewer's Responses to Questions

**Comments to the Author**

1. If the authors have adequately addressed your comments raised in a previous round of review and you feel that this manuscript is now acceptable for publication, you may indicate that here to bypass the “Comments to the Author” section, enter your conflict of interest statement in the “Confidential to Editor” section, and submit your "Accept" recommendation.

Reviewer #3: (No Response)

2. Is the manuscript technically sound, and do the data support the conclusions?

Reviewer #3: Yes

3. Has the statistical analysis been performed appropriately and rigorously? 

Reviewer #3: Yes

4. Have the authors made all data underlying the findings in their manuscript fully available?

Reviewer #3: Yes

5. Is the manuscript presented in an intelligible fashion and written in standard English?

Reviewer #3: Yes

6. Review Comments to the Author

Reviewer #3: The study evaluates the cost-effectiveness of COVID-19 booster doses and oral antivirals in the Indo-Pacific. However, the countries covered are Fiji, Indonesia, Papua New Guinea, and Timor-Leste. I am not sure exactly how big part from the Indo- Pacific they represent and would appreciate comment on this.

Generally, the paper is worth publishing, with a few suggestions for minor changes.

Minor. Methodologically, CEA, per se, is associated with well-acknowledged shortcomings, such as stochastic uncertainty, heterogeneity and treatment stratification, representation uncertainty, misinterpretation of censored data, list not exhaustive. Concerns regarding QALYs have been previously described in many articles, including a 2009 supplement to Value in Health. These limitations are not addressed in the relevant section of the paper.

Minor. Incremental net monetary benefit (INMB) can be presented as an alternative to the ICER. The INMB uses the WTP threshold to convert the QALY into a monetary value. Would the authors consider inclusion of INMB or at least comment its absence?

Minor. Authors are encouraged to elaborate a bit more about the “antivirals”. Indeed, Pfizer is mentioned /Paxlovid?/, are there other/s? It is important to avoid drugs subject to off- label use as well as to note the substantial price differences observed in the access provided during the pandemic period on a country-to-country basis.

Minor. In the PSA authors mentioned 1 000 times. Kindly rephrase to 1 000 Monte Carlo simulation, if this you meant.

Minor. I noticed self- citations in references. Authors are encouraged to explain and act on it.

7. PLOS authors have the option to publish the peer review history of their article (what does this mean?). If published, this will include your full peer review and any attached files.

Reviewer #3: **Yes: **Borislav Borissov

---

## [Author Response · Author response to Decision Letter 2]

11 Sep 2024

We thank the reviewers for their comments. These comments have improved the clarity of our methods. Please find details on how we have addressed reviewer comments in our response to reviewers document (2-pages).

---

## [Editor Report · Decision Letter 3]

16 Sep 2024

Cost-effectiveness analysis of COVID-19 booster doses and oral antivirals: case studies in the Indo-Pacific

PONE-D-23-34679R3

Dear Dr. Bilgin,

We’re pleased to inform you that your manuscript has been judged scientifically suitable for publication and will be formally accepted for publication once it meets all outstanding technical requirements.

Kind regards,

Dominic Luke Thorrington, PhD

Academic Editor

PLOS ONE

Additional Editor Comments (optional):

Thank you for your submission of the modified manuscript. The manuscript will now be accepted for publication.

---

## [Editor Report · Acceptance letter]

19 Sep 2024

PONE-D-23-34679R3 

PLOS ONE

Dear Dr. Bilgin, 

I'm pleased to inform you that your manuscript has been deemed suitable for publication in PLOS ONE. Congratulations! Your manuscript is now being handed over to our production team.

Kind regards, 

on behalf of

Dr Dominic Luke Thorrington 

Academic Editor

PLOS ONE